# iRGD synergizes with *PD-1* knockout immunotherapy by enhancing lymphocyte infiltration in gastric cancer

Naiqing Ding[1], Zhengyun Zou[1], Huizi Sha[1], Shu Su[1], Hanqing Qian[1], Fanyan Meng[1], Fangjun Chen[1], Shiyao Du[1], Shujuan Zhou[1], Hong Chen[1], Lianru Zhang[1], Ju Yang[1], Jia Wei[1] & Baorui Liu[1]

Poor infiltration of activated lymphocytes into tumors represents a fundamental factor limiting the therapeutic effect of adoptive cell immunotherapy. A tumor-penetrating peptide, iRGD, has been widely used to deliver drugs into tumor tissues. In this study, we demonstrate for the first time that iRGD could also facilitate the infiltration of lymphocytes in both 3D tumor spheroids and several xenograft mouse models. In addition, combining iRGD modification with *PD-1* knockout lymphocytes reveals a superior anti-tumor efficiency. Mechanistic studies demonstrate that the binding of iRGD to neuropilin-1 results in tyrosine phosphorylation of the endothelial barrier regulator VE-cadherin, which plays a role in the opening of endothelial cell contacts and the promotion of transendothelial lymphocyte migration. In summary, these results demonstrate that iRGD modification could promote tumor-specific lymphocyte infiltration, and thereby overcome the bottleneck associated with adoptive immune cell therapy in solid tumors.

---

[1] The Comprehensive Cancer Centre of Drum Tower Hospital, Medical School of Nanjing University & Clinical Cancer Institute of Nanjing University, 210008 Nanjing, China. These authors contributed equally: Naiqing Ding and Zhengyun Zou. Correspondence and requests for materials should be addressed to J.W. (email: weijia01627@hotmail.com) or to B.L. (email: baoruiliu@nju.edu.cn)

G astric cancer is a high-mortality disease with limited effective treatment options[1]. While recent developments in cell immunotherapy have already begun to revolutionize cancer treatment paradigms, the majority of patients with malignant solid tumors, such as gastric cancer, remain unresponsive[2]. Several pre-clinical and clinical studies have suggested a correlation between sufficient CD8[+] T cell infiltration and favorable prognosis[3,4]. However, studies have also demonstrated that less than 2% of transferred T cells actually infiltrate malignant solid tumors[5]. Aberrant adhesion molecule expression combined with heterogeneous tumor vessel permeability hinders lymphocyte extravasation[6]. Therefore, it is vital that this barrier be overcome to promote tumor-specific infiltration of lymphocytes[7].

It is a general concept that iRGD could function to promote extravasation and the tumor-specific penetration of small molecules and nanoparticles. The mechanism behind this process is thought to depend on the RGD domain and CendR motif. Specifically, the RGD sequence has been demonstrated to bind to ubiquitously expressed αvβ3 or αvβ5 in the tumor vascular endothelium and various tumor cells. These are then cleaved proteolytically by a cell-surface-associated protease, exposing the CendR motif. The truncated peptide loses its affinity for integrin

and binds to neuropilin-1 (NRP-1), triggering the penetration of compounds coupled to or co-delivered with it[8,9]. However, currently, no studies have been carried out to understand the effect of iRGD on lymphocyte infiltration. Based on this, we seek to explore whether modifying iRGD on T cell surface (T-iRGD) or co-delivering iRGD with T cells (T + iRGD) could also function to promote lymphocyte infiltration. We applied a time-efficient platform to connect iRGD to T cell surface and discovered that iRGD-modified T cells could penetrate into the core of the three-dimensional multicellular sphere while T cells alone could only gather on the edges of spheres. Meanwhile, iRGD modification could increase the number of T cells in the tumor parenchyma up to 10 times in different tumor modules in vivo. More importantly, iRGD modification synergizes with *PD-1* disruption in antitumor effect and prolonging survival in mouse model. Therefore, modifying T cells with iRGD may be an innovative strategy which would ultimately improve the therapeutic efficacy of adoptive cell therapy.

## Results

**Modification of T cells with DSPE-PEG-iRGD.** To immobilize iRGD on T cell membranes, we introduced a cysteine residue to

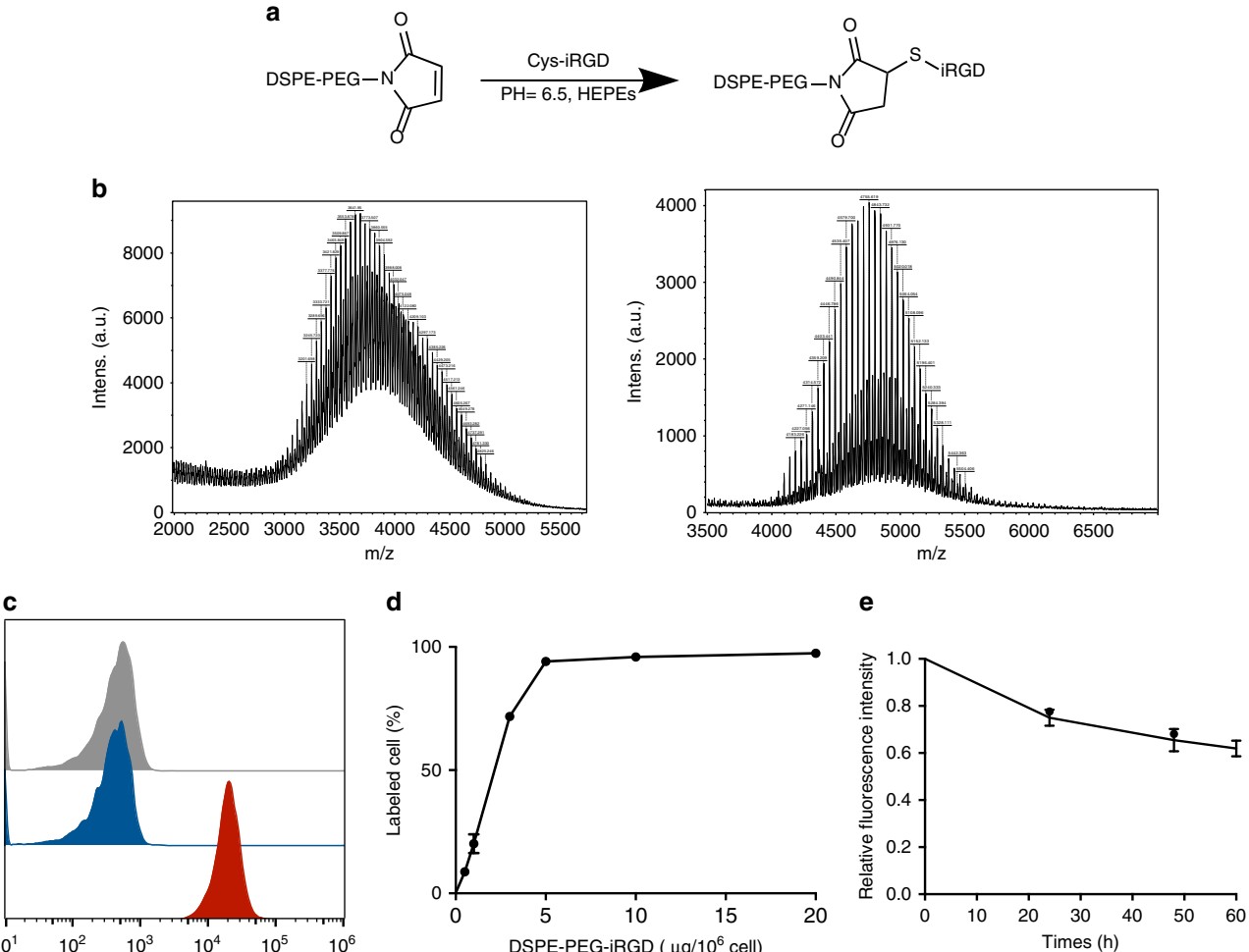

**Fig. 1** Synthesis of DSPE-PEG-iRGD and cell-surface modification with DSPE-PEG-iRGD. **a** Schematic diagram of the synthesis of lipid-conjugated iRGD. **b** MALDI-TOF characterization of DSPE-PEG-Mal and DSPE-PEG-iRGD construct. The difference in molecular weight indicates the successful connection of iRGD and DSPE-PEG-Mal. **c** Flow cytometry histograms of T cells alone (grey) and the cells incubated with iRGD-FAM (blue) and DSPE-PEG-iRGD-FAM (red). **d** Analysis of the percentage of DSPE-PEG-iRGD-FAM modified cell using flow cytometry. **e** Flow cytometric analysis of changes in relative averaged fluorescence intensities of cells modified with FAM-DSPE-PEG-iRGD over the culture periods. Data represent mean ± s.e.m.; n = 3. T + iRGD, T cells co-administered with free iRGD; T-iRGD, T cells modified with DSPE-PEG-iRGD

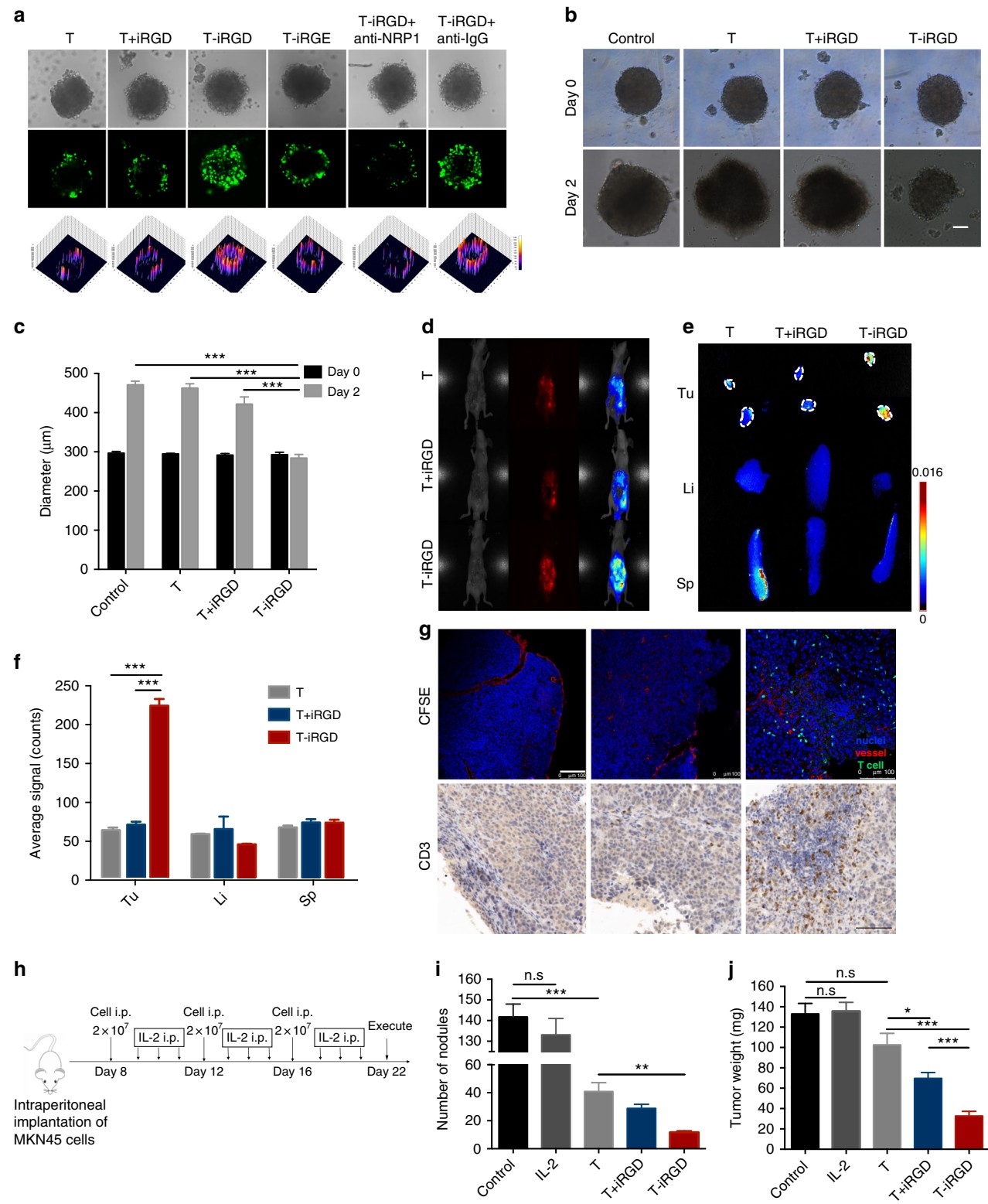

the C-terminal of the peptide. The free sulfhydryl group provided the potential to connect iRGD to the maleimide group of 2-distearoyl-sn-glycero-3-phospho-ethanolamine-N-maleimide (DSPE-PEG-Mal) through Michael addition reaction (Fig. 1a). MALDI-TOF and [1]H NMR analysis showed the successful production of DSPE-PEG-iRGD (Fig. 1b and Supplementary Fig. 1a). DSPE-PEG-iRGD-FAM was constructed using the same method for certain experiments. The resulting DSPE-PEG-iRGD-FAM

was showed to spontaneously transfer from solution to the T cell surface after co-culturing overnight (Fig. 1c and Supplementary Fig. 1b) without compromising the cell vitality, phenotype, or effector function (Supplementary Fig. 2a–e). In addition, 20 μg DSPE-PEG-iRGD created a 100% coating of $10^6$-activated T cells (Fig. 1d and Supplementary Fig. 1c). Because the binding stability is a critical parameter for cell-surface modification, we studied the cell-surface dynamics of DSPE-PEG-iRGD-FAM. The relative

**Fig. 2** T-iRGD showed improved tumor infiltration capacity and antitumor efficiency in 3D tumor spheroids and peritoneal metastasis tumor model. **a** Confocal microscopy images and surface plot images of HGC27 spheroids treated with CFSE-labeled T, T + iRGD, T-iRGD, or control peptide iRGE modified T cells (T-iRGE) for 20 h. 15 µg ml$^{-1}$ anti-NRP-1 antibody and control sheep anti-IgG were added 20 min before T cells incubation. Magnification, ×200; Scale bars, 100 µm. **b, c** T cell cytotoxicity against HGC27 tumor spheroids. Representative images of MCS treated with culture medium, T, T + iRGD, or T-iRGD for 20 h and then incubated in fresh culture medium for 24 h. scale bar, 100 µm (**b**). Growth inhibition assay in HGC27 MCS (**c**). Data are represented as mean ± s.e.m.; $n = 5$. Student's $t$ test. ***$p < 0.001$. **d** Near-infrared imaging of intraperitoneal tumor-bearing mice at 3 h post intraperitoneal injection of DiR-labeled T cells. **e**. Ex vivo images of tumor, liver, and spleen at 3 h after intraperitoneal T cells injection. White dashed lines, peritoneal tumors. **f** Semiquantification of T cells biodistribution at 3 h post-injection. Results are expressed as mean fluorescence intensity ± s.e.m. ($n = 3$). Student's $t$ test. ***$p < 0.001$. **g** Confocal images and immunohistochemistry analysis of resected tumor nodules at 3 h post intraperitoneal injection of T cells. Upper layer, T cells were labeled with CFSE before intraperitoneal injection; tumor blood vessels were stained with anti-CD31. T cell, green; vessel, red; nucleus, blue. Under layer, T cells were stained with anti-CD3. Scale bar, 100 µm. **h** Schematic illustration of treatment process of peritoneal metastasis tumor model. **i, j** Mice bearing disseminated MKN45 peritoneal tumors implanted 1 week earlier received intraperitoneal injection of PBS, IL-2, T, T + iRGD, or T-iRGD every 4 days for 3 times. Tumors are harvested after 2 weeks of treatment. Tumor nodules larger than 3 mm in diameter were weighed (**i**), and the numbers of remaining small tumor nodules (1–3 mm in general) were counted (**j**). Data represent mean ± s.e.m.; $n = 8$. Student's $t$ test. n.s, not significant; *$p < 0.5$; **$p < 0.01$; ***$p < 0.001$. Tu tumor, Li liver, Sp spleen

fluorescence intensity of DSPE-PEG-iRGD-FAM modified T cells declined to 50% after culturing for 60 h, which is approximately the doubling time of lymphocytes (Fig. 1e). This result suggested the favorable stability property of the cell-surface modification platform we have applied.

**T-iRGD possesses superior penetration capacity in MCSs.** Because 2D cultures do not work for cell infiltration studies, we generated three-dimensional multicellular sphere (MCS) using the gastric cancer cell line, HGC27 (Supplementary Fig. 3). The study demonstrated that lymphocytes alone hardly had the ability to penetrate into MCSs, with a weak signal detected only on the edges of the MCSs. Lymphocytes co-delivery with iRGD proved to be slightly more effective. Strikingly, iRGD modification resulted in lymphocytes being able to infiltrate deep into the MCS core in 20 h, nevertheless preinjection of an antibody to NRP-1 (anti-NRP-1) inhibited the iRGD-induced T cells penetration. The control peptide iRGE modified T cells also showed no tumor penetrating capacity (Fig. 2a). These results indicated that iRGD enhanced tumor-specific T cells accumulation is both RGD and CendR motif dependent. To test the cytotoxicity of the infiltrating T lymphocytes, we exposed HGC27 MCSs to different treatment groups for 20 h, followed by incubation in fresh culture medium for another 24 h. The tumor spheroid integrity was monitored and the mean diameter was measured by microscopy. Representative images indicate that neither T cells alone nor T cells co-delivered with iRGD inhibited MCS proliferation. However, we found that the MCS proliferation was totally suppressed and the spherical shape was almost destroyed with some dissociative cells distributed on the edge of MCS in the iRGD modification group (Fig. 2b, c). These ex vivo studies initially confirmed our hypothesis that iRGD endowed lymphocytes with a superior tumor infiltration capacity.

**iRGD facilitates T cells infiltration into peritoneal tumors.** The enhanced lymphocyte penetration into MCSs, which was found to be mediated by iRGD, encouraged us to study the tumor penetration capacity of T-iRGD in vivo. Peritoneal metastases are associated with a high morbidity and mortality in gastric cancer patients[10,11]. Thus, improving the current treatment options is urgently needed. Despite the superiority of lymphocytes local administration of lymphocytes in the treatment of peritoneal metastases, the efficiency of transferred lymphocytes remains restricted due to poor infiltration[12]. Previous studies have demonstrated that iRGD is advantageous for intraperitoneal drug accumulation through celiac infusion[13,14]. Therefore, we used a mouse model with MKN45 peritoneal metastatic gastric cancer to study whether iRGD is also involved in the abdominal tumor-

specific delivery of lymphocytes (Supplementary Fig. 3). Magnetic resonance imaging (MRI, 7 T) showed the successful establishment of peritoneal metastasis, similar to humans (Supplementary Fig. 4). Near-infrared imaging demonstrated that only the T-iRGD group exhibited the early accumulation of T cells (3 h post-injection) in small tumor nodules (Fig. 2d–f and Supplementary Fig. 5). Tumor sections from mice infused intraperitoneal injection with T-iRGD also showed enhanced infiltration of CD3$^+$ T cells compared to the other two groups (Fig. 2g). At 24 h post-transfusion, however, all the tested groups had different degrees of T cells accumulation in small tumor nodules (Supplementary Fig. 6a), whereas T-iRGD could even penetrate into large tumor nodules, and the average signal of the resected tumors was 2.8-fold to that of T cells alone (Supplementary Fig. 6b). Immuno-histochemical analysis confirmed the intensive tumor-specific infiltration of iRGD-modified lymphocytes: T cells, as detected by CD3 antibodies were found deep inside large peritoneal tumor nodules while the other two groups just showed T cells in the tumor periphery (Supplementary Fig. 6c). This indicated the superiority of iRGD-modified lymphocytes in regards to target tumor penetration in an intraperitoneal injection route.

We also studied the antitumor efficiency of T-iRGD cells in mice bearing disseminated MKN45 peritoneal tumors (Fig. 2h and Supplementary Fig. 7). All studied formulations exhibited strong cytotoxic activity against small peritoneal tumor nodules (<3 mm in diameter) relative to the untreated group, with T-iRGD cells found to be the most effective treatment method (Fig. 2i). In the case of larger tumor nodules with diameters >3 mm, lymphocytes alone exhibited a reduced tumor burden trend; however, the difference was not statistically significant ($p = 0.067$). Both T + iRGD and T-iRGD were found to result in significant tumor control, while the iRGD modification group was found to outperform iRGD co-delivery (Fig. 2j). None of the experimental groups exhibited alterations in body weight or liver and kidney function (Supplementary Fig. 8). This evidence demonstrated the favorable safety profile of regional T-iRGD administration. These results indicate that iRGD modification could function to enhance lymphocyte activity against not only small peritoneal tumor nodules but also bulky tumors when administered in an intraperitoneal injection route.

**iRGD promotes T cells accumulation in subcutaneous tumors.** We next evaluated the tumor-specific homing and extravasation of systemically delivered T-iRGD in an HGC27 subcutaneous tumor model. Whole-body fluorescence imaging demonstrated that, as expected, T cells alone distributed primarily in the liver and spleen, with no signal detected in the tumor area. The iRGD co-delivery group was found to be slightly more effective in

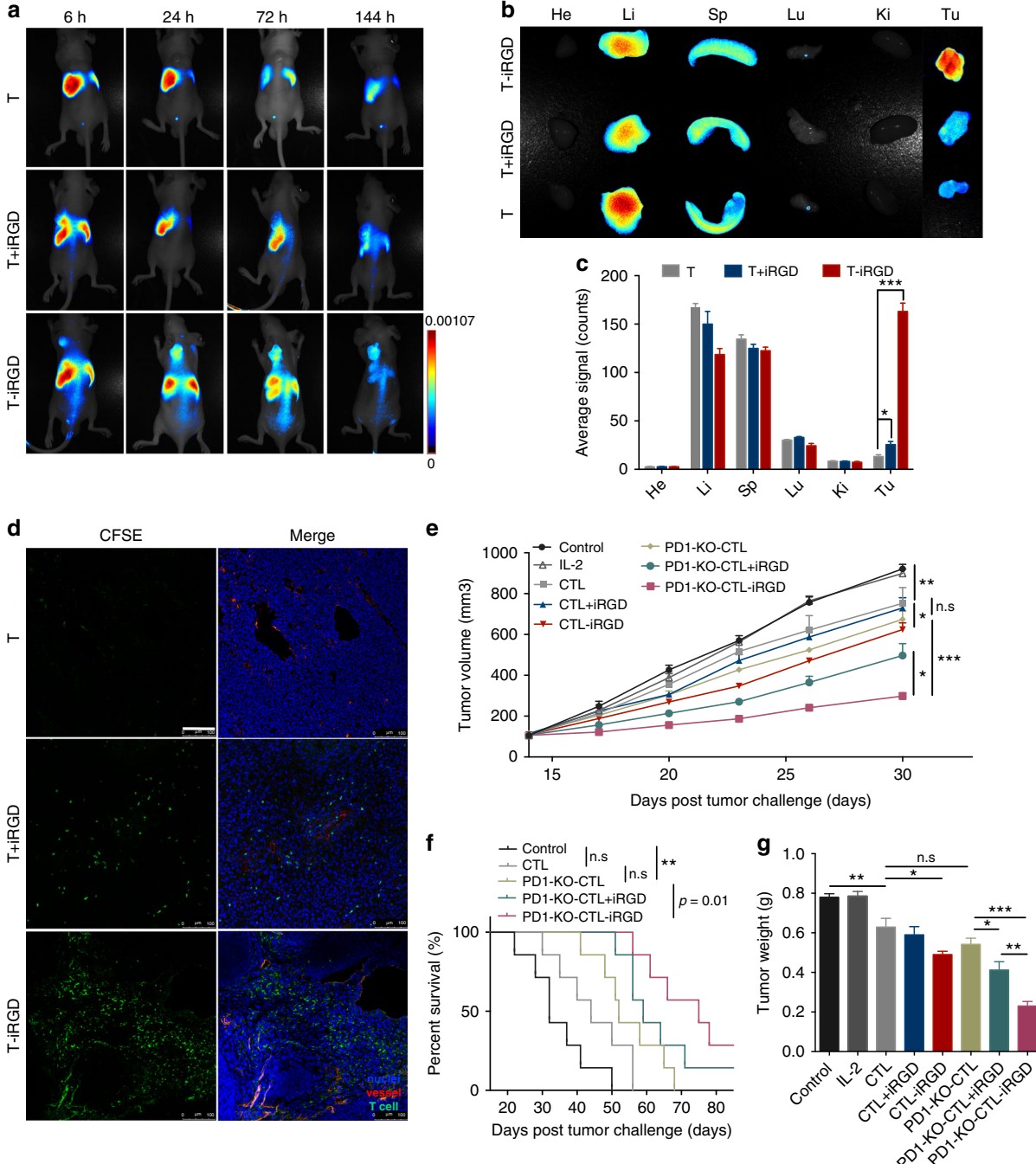

**Fig. 3** iRGD modification enhanced lymphocytes infiltration into tumor parenchyma in a systematic administration route and overcame resistance to *PD-1* disruption immunotherapy. **a** In vivo imaging of HGC27 tumor-bearing mice at 6, 24, 72, and 144 h after intravenous injection of DiR-labeled T cells. White dashed lines, subcutaneous tumors. **b** Ex vivo imaging of tumors and other organs at 144 h after T cells transfusion. **c** Semiquantification of T cells biodistribution in mice organs collected at 144 h post-injection. Results are expressed as mean fluorescence intensity ± s.e.m.; $n = 5$. Student's $t$ test. ***p* < 0.01, ****p* < 0.001. **d** Confocal imaging of frozen tumor sections at 24 h post T cells transfusion. T cells were labeled with CFSE before intravenous injection; tumor blood vessels were labeled with anti-CD31. T cells, green; vessel, red; nucleus, blue. Scale bar, 100 μm. **e–g** Enhanced antitumor effect of iRGD-modified *PD-1*-disrupted CTLs in a xenograft SNU719 mouse gastric tumor model. Tumor-bearing mice received different forms of treatment every 4 days for 3 times. Tumor growth profiles (**e**) and survival curve (**f**) of mice treated with PBS, IL-2, CTL, CTL + iRGD, CTL-iRGD, PD-KO-CTL, PD1-KO-CTL + iRGD, and PD1-KO-CTL-iRGD. Weight of tumors collected 2 weeks post treatment (**g**). Survival curves were analyzed with log-rank test. Tumor volume and tumor weight were analyzed with Student's $t$ test. Data are represented as mean ± s.e.m., $n = 7$; n.s, not significant; *$p < 0.05$; **$p < 0.01$; ***$p < 0.001$

tumor-targeted T cell penetration, with a faint signal observed in the tumor region. In T-iRGD group, however, T cells accumulated in the tumor as early as 6 h after transfer, increased by 24 h and maintained up to 144 h (Fig. 3a). At the final observation point, tissues were resected and measured ex vivo. Over 10-fold

more T cells were found within tumors in the T-iRGD group, with comparatively less accumulation in the liver and spleen (Fig. 3b, c). Confocal imaging of the resected tumors revealed a patchy distribution of T cells around the tumor blood vessels in the T + iRGD group, while iRGD-modified T cells exhibited a

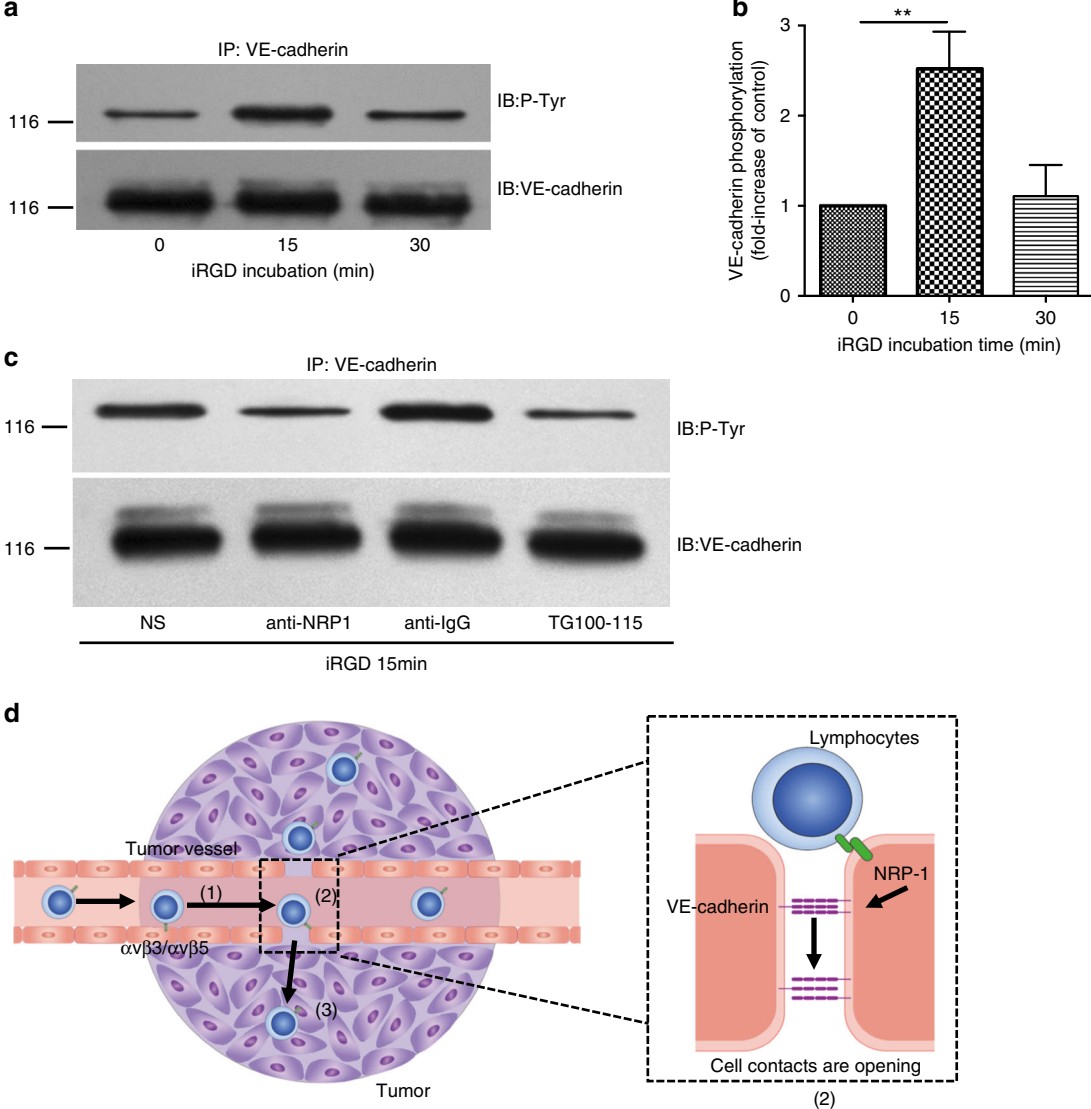

**Fig. 4** Potential mechanism of the effects exerted by iRGD on the extravasation of lymphocytes in tumors. **a** iRGD induced tyrosine phosphorylation of VE-cadherin in endothelial cells. Endothelial cells were incubated with iRGD and subjected to immunoprecipitation of VE-cadherin at the indicated time. The immunoprecipitate was then analyzed by immunoblotting using either anti-phospho-tyrosine or anti-VE-cadherin antibody. **b** The amount of tyrosine-phosphorylated VE-cadherin in panel (**a**) was quantified by densitometry from three independent experiments and expressed as fold-increase of untreated controls. Data represent mean ± s.e.m.; Student's $t$ test. ***$p < 0.001$. **c** Anti-NRP-1 antibody and PI3Kγ/δ inhibitor TG100–115 blocked iRGD-induced tyrosine phosphorylation of VE-cadherin. Endothelial cells were incubated with 15 μg ml$^{-1}$ anti-NRP-1 antibody, control sheep anti-IgG or 3.5 μg ml$^{-1}$ TG100-115 prior to iRGD incubation, tyrosine phosphorylation of VE-cadherin was analyzed by immunoprecipitation. **d** Schematic diagram of iRGD working mechanism. Circulating iRGD-modified T cells tether and roll in blood flow via the engagement of αvβ3/αvβ5 expressed on tumor vascular endothelial cell, slowing its velocity (**1**). The interaction also initiates the proteolysis of iRGD and expose the CendR motif. The truncated peptide then bind to NRP-1, triggering the tyrosine phosphorylation of VE-cadherin and the formation of intercellular gaps (**2**).The connected lymphocytes then cross the vessel wall and infiltrate into tumor parenchyma (**3**)

more extensive distribution in areas far from the blood vessels (Fig. 3d). Similar results were also observed in mice bearing SNU719, an EBV-positive gastric tumor model (Supplementary Figs 3 and Supplementary Fig. 9a–d).

**iRGD modification synergizes with *PD-1* disruption**. EBV-associated gastric cancer is a type of gastric tumor that was recently defined according to a novel classification system[15]. EBV-associated gastric cancer is characterized by high *PD-L1* expression level and abundant lymphocyte infiltration and is believed to be sensitive to immune checkpoint disruption therapy[16]. In EBV-associated gastric cancer, LMP2A is an ideal target for adoptive cell immunotherapy[17,18]. The *PD-1* expression level

in EBV-LMP2A-CTLs has been altered using a CRISPR-Cas9 system and the *PD-1* knockout lymphocytes showed enhanced cytotoxicity against an EBV-positive gastric cancer cell line in vitro. Whereas the in vivo antitumor efficiency was found to be limited, which is attributed to poor infiltration of the infused lymphocytes[19]. Thus, combining iRGD with *PD-1* disrupted lymphocytes adoptive therapy (PD1-KO-CTL) represents a promising strategy. EBV-LAM2A-positive CTLs and PD-1-disrupted EBV-LMP2A-CTLs (PD1-KO-CTL) were generated as described previously. The antitumor effect was studied in mice bearing SNU719 xenograft tumor model. These studies demonstrated that CTL-iRGD resulted in increased tumor control compared to CTLs, further confirming the superiority of CTL-iRGD in the

systemic administration route. Still, the antitumor effect remained weak, which indicated that enhanced T cell infiltration alone may not be sufficient to control solid tumor growth. In addition, the *PD-1* disruption was found to cause tumor regression and prolong survival to a certain extent. However, consistent with our previous studies, the antitumor effect was not any better compared to the CTL group. As expected, iRGD-modified PD1-KO-CTL elicited maximum antitumor efficacy in both tumor control and prolonging survival (Fig. 3e–g). No body weight alterations were observed in any studied groups (Supplementary Fig. 10). Thus, these results implied that the immune-suppressive pathway functioned to hamper the infiltrated lymphocytes' ability to destroy tumor cells. More importantly, sufficient lymphocyte infiltration is a required prerequisite for the response to checkpoint blockade immunotherapy. These studies demonstrated that iRGD modification combined with *PD-1* disruption possesses a far-reaching therapeutic efficiency.

**iRGD induces tyrosine phosphorylation of VE-cadherin**. Due to the confirmed role of iRGD modification in the promotion of lymphocyte infiltration in the solid tumor parenchyma, we aimed to understand its underlying molecular mechanism. Studies have demonstrated that iRGD binding to NRP-1 could function to induce the formation of grape-like vesicles in the cytoplasm[20]. This is thought to mediate the transcytosis of small molecules, monoclonal antibodies, and nanoparticles. In addition, these newly defined cellular organelles, termed the vesiculo-vacuolar organelle (VVO), have been reported to play a role in the formation of cytoplasmic channels and the transcellular migration of lymphocytes[21]. Thus, we inferred that iRGD may function to facilitate lymphocyte transcellular migration through the formation of VVO. In addition to transcellular migration, lymphocyte transendothelial migration relies heavily on a paracellular mode[22]. Numerous studies have demonstrated that the vascular endothelial adhesion molecule, VE-cadherin, regulates the stability of endothelial junctions and participates in the process of lymphocyte transendothelial migration[22,23]. Tyrosine phosphorylation of VE-cadherin has been demonstrated to induce the disengagement of endothelial junctions and the formation of intercellular gaps, and thus, transendothelial lymphocyte migration[23]. For NRP-1 activation has been reported to induce tyrosine phosphorylation of VE-cadherin in a PI3Kγ/δ dependent manner[24], we aimed to understand whether the mechanism also plays a role in iRGD-induced lymphocyte infiltration. In endothelial cells, iRGD treatment was found to result in a rapid and transient increase in VE-cadherin tyrosine phosphorylation (Fig. 4a, b). To extend these studies, endothelium cells were pretreated with a function-blocking NRP-1 antibody or the PI3Kγ/δ inhibitor TG100-115 before iRGD incubation. As expected, both NRP-1 antibody and PI3Kγ/δ inhibitor potently inhibited iRGD-induced tyrosine phosphorylation of VE-cadherin (Fig. 4c). This research verified the underlying mechanism of iRGD in promoting lymphocyte infiltration that the binding of iRGD to NRP-1 triggered the formation of intercellular gaps through the tyrosine phosphorylation of VE-cadherin (Fig. 4d).

## Discussion

iRGD is a cyclic tumor-specific homing and penetrating peptide identified by phage display technique[8]. The peptide has been widely studied for tumor-specific delivery of small molecules, nanoparticles, and mono-antibodies. The mechanism of action lies in the induction of a vast transcytosis pathway mediated by NRP-1. One major advantage of this peptide is that the tumor-specific penetrating capacity does not rely on the chemically conjugated to the carrier[9]. In this study, we explored a new

application field of iRGD in adoptive cell immunotherapy for the first time. We have also discovered a novel functional mechanism of iRGD that differs from its previously reported vascular transcytosis role in the transportation of drugs and nanoparticles. In our studies, unconjugated iRGD was found to exhibit limited effects in lymphocyte transport. This was at odds with the transportation of small molecules. It is possible that this could be caused by the short opening time of the intercellular gaps, as well as the different trafficking and infiltrating times between the small molecules and the cells. Thus, a continuous infusion of iRGD following lymphocyte transfusion may be a more appropriate approach compared to co-administration.

T cells play a pivotal role in the immune responses against cancer whereas the fully functional T cells generated ex vivo are often hampered by the inability to infiltrate into the tumor site[2]. The process of circulating T cells migrate to the tissues includes tethering, rolling, adhesion, and transmigration. For the majority of solid tumors, mismatching of chemokine/chemokine receptors, abnormal angiogenesis, downregulation of adhesion molecules, and tumor immune suppression mechanisms restrict the infiltration of T cells[7,25]. Strategies like chemokine receptor engineering, anti-angiogenesis, or administering checkpoint inhibitors have been proposed[25]. Here we adopted a lipid-insertion approach to functionalize T cells with the tumor penetrating peptide iRGD. This approach allowed for the rapid modification of T cells with iRGD without complicated genetic manipulation and can enhance tumor-specific T cells infiltration from many aspects. Firstly, the binding of RGD domain to αvβ3/αvβ5 expressed in the tumor vascular endothelium increases the leukocyte-vessel wall interaction, thereby favoring the tethering and rolling of T cells on endothelium. Secondly, the activation of NRP-1 loses VE-cadherin dependent adherence junctions, increasing vascular permeability and promoting T cells extravasation. Besides, the transient increase in vascular permeability also facilitates the penetration of chemotherapeutic agents which have been shown to promote intratumoral expression of chemokines attracting T cells[26]. Still, the trafficking of lymphocytes alone may not be entirely sufficient for successful immunotherapy. Needless to say, combining with other therapeutic strategies, such as local radiotherapy, chemotherapy, or other immunotherapies could be perceived as pivotal for further increasing therapeutic success.

## Methods

**Reagents**. The cyclic peptide Ac-CCRGDKGPDC-NH2 (C-iRGD) containing an extra cysteine at the C-terminus of iRGD, the control peptide Ac-CCRGEKGPDC-NH2 (C-iRGE), as well as iRGD with FAM conjugated at the N-terminus were purchased from (Top-Peptide, China). 1,2-Distearoyl-sn-Glycero-3-Phosphoethanolamine-N-[Maleimide(polyethylene-Glycol)-3400] (DSPE-PEG-Mal) was purchased from (Laysan Bio, Inc, USA).

**Cell lines**. Human gastric adenocarcinoma cell line MKN45, SNU719, HGC27 were purchased from the Cell Bank of Shanghai Institute of Biochemistry and Cell Biology before use, cultured in RPMI 1640 medium supplemented with 10% fetal calf serum, 100 U mL$^{-1}$ penicillin and 100 μg mL$^{-1}$ streptomycin. Human umbilical vein endothelial cells (HUVECs) were cultured in EBM-2 medium (Lonza, USA). All cells were incubated at 37 °C and 5% CO$_2$, authenticated by checking morphology by microscopy after plating at different concentrations. Cells were tested for Mycoplasma and only Mycoplasma free cells were used.

**Synthesis of DSPE-PEG-iRGD**. DSPE-PEG-Mal was mixed with C-iRGD or C-iRGD-FAM at a 1:1 molar ratio in Hepes buffer (pH = 6.5). This reaction mixture was gently stirred at room temperature for 48 h under nitrogen gas. After that, the resulting reaction mixture was placed in a dialysis bag (molecular weight cutoff = 3500 Da) and dialyzed in deionized water for 48 h to remove the free iRGD. The final solution in the dialysis bag was lyophilized and analyzed by matrix-assisted laser desorption/ionization-time-of-flight-mass spectrometry (MALDI-TOF MS) and $^1$H NMR spectroscopy. DSPE-PEG-iRGE was synthesized using the same method.

**Isolation and culture of primary human T lymphocytes.** The blood collection procedure was carried out in accordance with the guidelines verified and approved by the Ethics Committee of Drum Tower Hospital. All donors signed an informed consent for scientific research statement. Peripheral blood mononuclear cells (PBMCs) were isolated from samples from healthy volunteers by centrifugation on a Ficoll density gradient and suspended in AIM-V medium (Gibico, USA). PBMC were cultured by adherence for two hours and non-adherent T lymphocytes were activated with 1000 U ml$^{-1}$ IFN-γ (PeproTech, USA) on day 1 and 50 ng ml$^{-1}$ OKT3 (eBioscience, USA) on day 2. Cells were expanded in complete medium containing 90% AIM-V (Gibico, USA), 10% FBS serum (Gibico, USA), 300 U ml$^{-1}$ IL-2 (Peprotech, USA) and 50 ng ml$^{-1}$ IL-15 (PeproTech, USA). For the generation of *PD-1* disrupted T cells, OKT-3 activated T cells were transfected with the intended plasmids by Nucleofector 2B (Lonza, Germany). 10$^7$ cells were washed with DPBS and resuspended in 100 μl transfection buffer (Amaxa Human T cells Nucleofector Kit, VPA-1002, Lonza, Germany). Program T-007 was selected. After transfection, cells were resuspended in 500 μl pre-warmed AIM-V medium and cell culture medium was half replaced by fresh complete medium containing 100 U ml$^{-1}$ IL-2 every 2–3 days[19].

To generate EBV-LMP2A-CTLs and *PD-1* disrupted EBV-LMP2A-CTLs, adhered DCs were cultured in AIM-V medium containing 500 U ml$^{-1}$ human IL-4 (PeproTech, USA) and 500 U ml$^{-1}$ GM-CSF (PeproTech, USA) for four days. On day 5, fresh complete medium containing 500 U mL$^{-1}$ TNF-α (PeproTech, USA), 500 U ml$^{-1}$ IFN-α (PeproTech, USA), and 50 ng ml$^{-1}$ PGE2 (PeproTech, USA) was added and the culture was continued for two days. On day 7, mature DCs were pulsed by LMP2A peptide at the concentration of 10 mg ml$^{-1}$ for 4–6 h at 37 °C and then incubated with activated T cells or *PD-1* disrupted T cells at a ratio of 1:10 in complete AIM-V medium supplemented with 25 ng ml$^{-1}$ IL-7 and 10 ng ml$^{-1}$ IL-15. Fresh complete medium containing cytokines (100 U ml$^{-1}$ IL-2, 10 ng ml$^{-1}$ IL-7, and 10 ng ml$^{-1}$ IL-15) was added every 2–3 days until use for experiments[19].

**Cell-surface modification with DSPE-PEG-iRGD.** Activated T cells were seeded at the concentration of 1–2 × 10$^6$ ml$^{-1}$ one day prior to iRGD coating. DSPE-PEG-iRGD, DSPE-PEG-iRGE, or DSPE-PEG-iRGD-FAM was added into cell culture medium at the indicated concentration and incubated for 30 min in 37 °C at 5% CO$_2$ incubator to achieve the modification of iRGD on the cell surface. Cell morphology was observed under optical microscope post iRGD modification. The cells were then washed with PBS and centrifuged (180× *g*, 5 min, RT) to remove unreacted reagents. The ratio of DSPE-PEG-iRGD-FAM modified T cells was analyzed by flow cytometry. iRGD modification on EBV-LMP2A-CTLs and *PD-1* disrupted EBV-LMP2A-CTLs was achieved using the same method. Cell vitality was assessed by PI staining. The cytotoxicity of iRGD-modified EBV-LMP2A-CTLs were assessed by CFSE/PI labeling cytotoxicity assay.

**Stability of iRGD conjugates on cell surface.** DSPE-PEG-iRGD-FAM was immobilized on T cell surface at saturation quantity as mentioned above. To examine the stability of the iRGD conjugates immobilized on the cell surfaces of T cells, iRGD-modified T cells were incubated in the complete medium in 37 °C at 5% CO$_2$ incubator. At indicated time intervals, the cells were collected by centrifugation (180 × *g*, 5 min, RT) and analyzed by flow cytometry.

**Flow cytometry.** We performed flow cytometry analysis using the following antibodies: CD3-Percp-Cy5.5 (HIT3a, BD Bioscience), CD8-APC (RPA-T8, BD Bioscience), CD27-PE (M-T271, BD Bioscience), CD28-APC (CD28.2, BD Bioscience), CD45RO-PE (UCHL1, BD Bioscience), CD62L-FITC (DRGE-56, BD Bioscience), αvβ3-FITC (LM6090, EMD Millipore,), αvβ5-FITC (P1F6, EMD Millipore), and NRP-1-PE (AD5-17F6, Miltenyi Biotec GmbH). All samples tested were suspended in FACS buffer and stained with indicated antibodies for 30 min in 4 °C in darks, and then, washed twice, and resuspended in FACS buffer before analysis. For IFN-γ detection, BD™ Cytometric Bead Array (CBA) Human IFN-γ kit (BD Bioscience, USA) was used. Samples were analyzed with BD Accuri C6 (BD Bioscience).

**T cell experiments in MCSs.** The HGC27 cells (500 in 150 μl of complete media) were added to 96 Well Clear Round Bottom Ultra Low Attachment Microplate (Corning, USA) and allowed to grow up at 37 °C to attain the diameter about 200 μm for 72 h. MCSs were monitored with a microscope and the uniform and compact tumour spheroids were selected for the subsequent studies. To study the MCSs penetration of T lymphocytes, T cells were stained with 4 μM CFSE (Carboxyfluorescein succinimidyl ester) (Abcam, UK) for 10 min at 37 °C in PBS before iRGD modification. Labeling was stopped by 5-fold volume of cold complete medium (10% FBS in AIM-V) and extensively washing for three times. Established spheroids were first incubated with 15 μg ml$^{-1}$ function-blocking anti-NRP-1 antibody (R&D Systems) or control sheep anti-IgG (R&D systems) for 20 min at 4 °C. CFSF-labeled T cells, T cells combined with iRGD, T cells modified with DSPE-PEG-iRGD or T cells modified with DSPE-PEG-iRGE were then added to the solution and incubated with the spheroid for 20 h. After washing and fixing in 4% paraformaldehyde, tumour spheroids were scanned from the top to the middle with 10 μm intervals using confocal microscope (ZEISS, Germany) and the images were acquired at the mid-height of the spheroid. The images were also analyzed

using ImageJ software. Cytotoxicity experiments in MCSs were conducted by exposing MCSs to activated T cells, activated T cells combined with iRGD or activated T cells modified with DSPE-PEG-iRGD at the E: T ratios of 5:1 for 20 h and then incubated in fresh culture medium for another 24 h. The spheroid diameter and morphological change were monitored by bright field microscopy (Leica, Germany).

**Xenogeneic mouse models.** The Ethics Committee of Drum Tower Hospital approved all experiments in this study. All animal procedures were carried out in compliance with guidelines set by the Animal Care Committee at Drum Tower Hospital (Nanjing, China). Investigators were not blinded for animal studies. All efforts were made to minimize the number of animals used and their suffering. Mice were randomized on the basis of age and weight.

For peritoneal metastasis tumor model, 6–8-week-old male BALB/c nude mice were injected intraperitoneally with 10$^6$ MKN45 cells. The formation of peritoneal metastasis tumor nodules was assessed by T2-weighted 7.0 T micro-MRI (PharmaScan, Germany).

For subcutaneous tumor model, 6–8-week-old male BALB/c nude mice were injected subcutaneously with HLA-A24 positive SNU719 cells (1 × 10$^7$ suspended in 100 μl PBS) or HGC27 cells (5 × 10$^6$ suspended in 100 μl PBS).

**In vivo near-infrared fluorescence imaging.** To investigate the tumor targeting efficiency of T-iRGD in tumor-bearing mice, 10$^7$ T cells stained with near-infrared fluorescent probe DiR (Bridgen, China) were injected intraperitoneally (MKN45 peritoneal metastasis tumor model) or intravenously (HGC27 and SNU719 subcutaneous tumor model). At different time intervals, the mice were anesthetized and scanned using a CRi Maestro™ Automated In Vivo Imaging System (C.R. INTERNATIONAL INC, USA). In some experiments, 50 μg of function-blocking anti-NRP-1 antibody or sheep IgG was intravenously injected into the tumor mice 15 min prior to the T cells injections. At the end of the observation time point, the mice were sacrificed, and the tumor and major organs were resected and imaged.

**Immunofluorescence microscopy imaging.** Immunofluorescence staining for activated T cells in the tumors was analyzed using confocal microscope (Leica, Germany). 10$^7$ T cells were stained with CFSE (Abcam, UK) before intravenous (subcutaneous tumor model) or intraperitoneal (peritoneal metastasis tumor model) injection. Tumors were harvested at indicated time point and processed for immunostaining. Frozen sections were stained with CD31 using rabbit anti-mouse CD31 antibody (Abcam, UK) followed by the Cy3-conjugated goat anti-rabbit IgG (Abcam, UK) secondary antibody. After washing with PBS, the sections were mounted with DAPI (Beyotime, China) and analyzed at ×200. A minimum of four fields for each tumor section were analyzed.

**immunohistochemistry analysis.** For immunohistochemistry staining, 10$^7$ T cells were injected intraperitoneally in peritoneal metastasis tumor model. Tumors were harvested at indicated time point and fixed in 10% neutral-buffered formalin and embedded in paraffin. Infiltrated human CD3$^+$ T cells were detected by rabbit anti-human CD3 mAb SP7 (Abcam, UK).

**In vivo antitumor efficacy.** For the MKN45 peritoneal metastasis tumor treatment study, tumor-bearing mice were randomized in four groups (*n* = 8) and treated every four days with IP injection of 0.3 ml PBS, T, T + iRGD, or T-iRGD for three times. 40,000 U human recombinant IL-2 were given intraperitoneally once a day for three consecutive days after T cells transfer. Mice were weighed every two days. Two weeks after the start of treatment, peripheral blood serum was collected for assessment of kidney and liver function. Mice were sacrificed, the tumors and organs were excised. To estimate the tumor burden, large tumor nodules (tumor bigger than 3 mm in diameter) were weighted and small tumor nodules (tumor smaller than 3 mm in diameter) were counted. One mouse from each group was randomly selected and main organs were collected for histology analysis. Organs were fixed in 10% neutral-buffered formalin, embedded in paraffin, sliced, and stained with hematoxylin-eosin (H&E).

In SNU719 treatment study, subcutaneous tumor-bearing mice were randomized into seven groups (*n* = 7). Two weeks post tumor induction, mice were treated with intravenous injection of 0.1 ml PBS, CTL, CTL + iRGD, CTL-iRGD, PD1-KO-CTL, PD1-KO-CTL + iRGD or PD1-KO-CTL-iRGD, respectively, every four days for a total of three times. 40,000 U human recombinant IL-2 were given intraperitoneally once a day for three consecutive days after T cells transfer. At the time of adoptive T cell transfer, mice were weighed and inspected for tumor development. The tumor size was determined using a caliper. By assuming an ellipsoid tumor shape, the volume was calculated as length × width$^2$ × 0.5.

**Immunoprecipitation of VE-cadherin.** Confluent HUVECs (p 4–6) were starved in serum-free EBM-2 media (Lonza, USA) for 16 h prior to 20 min incubation with 15 μg ml$^{-1}$ function-blocking anti-NRP-1 antibody, control sheep anti-IgG or 3.5 μg ml$^{-1}$ TG100-115. Then, cells were stimulated with iRGD (500 ng mL$^{-1}$) for 15-min or 30-min. To immunoprecipitate VE-cadherin[24], cells were harvested and

lysed using the cell lysis buffer (KeyGEN BioTECH, China). Protein concentrations of cell extracts were determined using the BCA protein assay (Pierce, USA) and VE-cadherin was immunoprecipitated from cell extract using anti-VE-cadherin antibody (D87F2, Cell Signaling Technology, USA) and Protein A Magnetic Beads (Cell Signaling Technology, USA). Immunoblotting of precipitated protein was carried out using antibodies against VE-cadherin (Cell Signaling Technology, USA) and phosphotyrosine (4G10, Millipore, USA). Uncropped images of blots and gels are shown in the Source Data file.

**Statistical analysis**. Graphpad Prism 5.0 (Graphpad software, San Diego, CA) and SPSS were used for all statistical analysis. Variance was similar between the groups that were compared statistically. No statistical methods were used to predetermine sample size. Data are presented as mean ± s.e.m. unless indicated otherwise. $P < 0.05$ was considered statistically significant.

**Reporting summary**. Further information on experimental design is available in the Nature Research Reporting Summary linked to this article.

## Data availability

The source data underlying Figs 1d, e, 2c, f, i, j, 3c, e, g, 4a–c and Supplementary Figs 1b, c, 2b–d, 6b, 8a, 8c, 9c, 10 are provided as a Source Data file. The data that support the findings of this study are available within the article and its supplementary information files and from the corresponding author upon reasonable request. A reporting summary for this article is available as a Supplementary Information file.

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

## Acknowledgements

This work was funded by grants from the National Natural Science Foundation of China (No. 81672367 and 81502037), National Key Research and Development Program of China (grant number 2017YFC1308900, 2018ZX09301048-003), Scientific and Technological Development Program of Jiangsu Province (No. BE2017607). The funding sources had no role in the study design, data collection, data analysis, data interpretation, or writing of the report.

## Author contribution

N.D. conducted experiment and wrote the main manuscript text and prepare figures, N. D., Z.Z., B.L., and J.W. conceived and designed all experiments. S.S., F.M., H.S., and F.C. provided protocol and tools for research. H.Q., L.Z., S.Z., H.C., and S.D. assisted in conducting experiment and prepare figures. S.S., J.Y., and H.Q. assisted in manuscript preparation. B.L. and J.W. designed & coordinated research, verified results. All authors reviewed the manuscript.
