## [Peer Review File · Nature Communications]

Reviewers' Comments:

Reviewer #1:

Remarks to the Author:

This is an interesting paper with a potentially important message that relates to the highly topical field of tumor immunotherapy. The authors show that accumulation of T lymphocytes in tumors can be enhanced by coating the cells with a tumor-homing peptide (iRGD), and to an extent by co-administering the cells with the peptide. This peptide has tumor-penetrating properties, and it has been shown to promote the delivery of various types of drugs to tumors, but it has not been previously used to do the same with cells. The authors also show in a peritoneal carcinomatosis model that the iRGD-coated lymphocytes enhance the anti-tumor efficacy of PD-1-directed anti-cancer therapy.

The results seem solid as far as they go, but an important control is missing, and one set of the results is incomplete and provides insufficient support to conclusions drawn from it.

1. The authors use unmodified T cells as a control for the iRGD-coated cells. This comparison does not provide formal proof for the conclusion, however likely, that iRGD is responsible for the difference in the tumor accumulation of the two types of cells. The cells also differ with regard to the DSPE-PEG moiety that attaches the iRGD peptide to the T cell surface. Control cells, in which an inert peptide has been similarly attached to T cells is needed. They should also inhibit the tumor penetration with anti-NRP-1 antibodies.

2. The last section on the mechanism of iRGD-induced T cell penetration is woefully incomplete and fails to adequately support the conclusion presented in the title of the section, which reads: "The enhanced tumor-specific infiltration of iRGD-modified T cells depends on the tyrosine phosphorylation of VE-cadherin." Only a correlation is shown between the infiltration and V-cadherin phosphorylation. Minimally, experiments such as inhibiting the phosphorylation with anti-NRP-1 and using endothelial cells with a mutant V-cadherin that cannot be phosphorylated would be needed. Direct visualization of lymphocyte penetration through tumor endothelium by EM, as was done in reference 23 with nanoparticles, would be another way of demonstrating the authors claim.

There are some writing-related issues that could cause confusion:

'Aggregation' is used in the meaning of 'accumulation' in describing accumulation of T cells in tumors. As aggregation generally means clustering in the context of cells, this usage is confusing.

"Immunohistochemical analysis confirmed the intensive tumor-specific infiltration of iRGD-modified lymphocytes: CD3+ T cells were found deep inside large peritoneal tumor nodules ..." A reader wonders why T cells now became CD3+ T cells. The figure legend explains that the antibody used to detect T cells was anti-CD3. "T cells, as detected by CD3 antibodies" would take care of it. 'Systematically' should be 'systemically', 'merely' 'only', and 'cycled' 'cyclic'.

Reviewer #2:

Remarks to the Author:

In the submitted manuscript, the authors demonstrate that loading the tumor penetrating cyclic peptide iRGD on the cell surface of T cells enhances their ability to penetrate tumors in vitro and in vivo. Some of the provided imaging data is intriguing, however the manuscript lacks detailed mechanistic studies. I have several major concerns that are summarized below.

Major concerns:

1) The T-cell accumulation data shown in Fig 3a-c is intriguing. However, analysis is stopped after 6 days. Please provide longer follow up.

2) The animal experiments testing the antitumor activity of iRGD-loaded T cells are confounded by the administration of multiple doses of very high doses of IL2 (40,000 units). In this regard, it is critical to have a group of mice as controls that only receive IL2.

3) It is unclear why the investigators use unspecific T cells for the MKN49 experiments. In this regard please provide data that these allogeneic T cells recognize and kill MKN49 cells (right now only growth inhibition of spheroids is shown in Fig 2c). Also, for the 2nd model, please provide data that the injected LMP2-specific T cells recognize and kill SUN719 cells ex vivo; while the authors have published this before, generating these cells is in part donor dependent. Lastly, I could not find the HLA type of SUN719 cells in the literature – please provide.

4) No complete tumor regressions were observed in the animal models. In addition, the shown survival benefit in the 2nd animal model seems marginal (no stat analysis was provided). Thus, believe that it is critical that the authors provide robust mechanistic studies. The studies shown in Fig 4 have to be considered preliminary.

Minor:

1) Please provide stat analysis for the survival curve shown in Fig 3f.

Reviewer #3:

Remarks to the Author:

The manuscript "iRGD synergizes with PD-1 knockout immunotherapy by enhancing lymphocyte infiltration in gastric cancer" proposes a new strategy for oncological cell immunotherapy. The authors show that iRGD-modified lymphocytes exert efficient infiltration to tumor mass in 3D cell culture model and in mouse tumour models. The iRGD-modified lymphocytes also demonstrated efficient anti tumour activity in 2 mouse tumour models. Finally, PD-1 knockout in the lymphocytes enhanced the anti-tumour efficiency even further.

The manuscript is generally well written, except for some misspellings in the text. Strong result is presented – a method for effective antitumor treatment, which has been demonstrated in 2 different mouse models. The methodology is adequate (assuming that the questions listed subsequently will be sufficiently answered), as is the analysis and presentation of data. The manuscript should be published if the questions will be sufficiently answered.

Major

The whole-body scan images do not contain appropriate scale, which needs to be fixed by the authors. The authors have presented just a color ribbon, with no numerical reference to it. Secondly, in live animal imaging studies (Fig 3a and ExtendedData7a) one can see that as time passes (from 6h -> 144h), the overall signal in the whole animal scan is increasing with time, while one would assume that the signal decreases in time. Are the different time points (6h vs 24h vs 72h vs 144h) in the same overall scale? Or does a different scale apply for each time point? How do the authors explain that as time passes, the amount of their injected material (T-cells injected at time 0h) is increasing? In the ExtendedData7a "T-iRGD" group it even seems that the T-cells are labelling up the whole body (including the head region, which the authors do not comment).

Fig 3a live animal scans in T-iRGD group contain a strong signal origin that the authors do not comment on. It is impossible for the reader to interpret the signal (the images are small and of low quality), but it seems that one side/part of the lungs gets infiltrated with the lymphocytes?

Similarly, as mentioned before, in ExtendedData7a at 144h one can see very strong signal in the head region.

Minor and technical remarks

Language

- L99 one instance of "lymphocytes" should be removed.
- In the whole manuscript, there are several instances where "systematic injection" should be replaced with "systemic injection".

Illegible Figures

- There are illegible (too small) elements in several figures: for example, Fig 1b, Fig 2h.
- Extended Fig1a – the top is missing

Replicability: the authors should make sure their study can be replicated by the other scientists:

- The monitoring and measurements of MCS (spheres). L406 "MCS ... were used for functional assays upon reaching a solid state approximately 72 hours" – the size of the spheroids should be indicated. When presenting the results of the infiltration of labelled T-cells into the MCS, L414 "...T cells in MCS were monitored using confocal microscope", but the authors do not mention or describe how they ensured the comparability between the different spheroids. Usually the imaging and quantitation is performed at the mid-height of the equal-sized spheroids.

Reviewer #1:

Comment 1:

The authors use unmodified T cells as a control for the iRGD-coated cells. This comparison does not provide formal proof for the conclusion. Control cells, in which an inert peptide has been similarly attached to T cells is needed. They should also inhibit the tumor penetration with anti-NRP1 antibodies.

Answer:

Thank you very much for your comments. We have added three groups (T-iRGE, T-iRGD+anti-NRP1, T-iRGD+anti-IgG) in our experiments to make a more reliable conclusion.

In 2009, Sugahara et al. published their work on *Cancer Cell*, in which they reported iRGD as a cyclic peptide with tumor specific penetration capacity. iRGD demonstrated the ability of homing to tumors through three-step process: the RGD motif mediates binding to $\alpha\beta3/\alpha\beta5$ integrins on tumor endothelium and a proteolytic cleavage then exposes a binding motif for neuropilin-1(NRP-1), which mediates penetration into tissue and cells. Besides, they also found that control peptides including a non-integrin binding variant, CRGEKGPDC (iRGE) has no tumor homing capacity. In addition, anti-NRP1 antibody also blocked tumor penetration of iRGD. Therefore, the in vivo homing function of iRGD to tumors is dependent on both the RGD and CendR motif[1].

According to this study, we have synthesized an inert peptide CCRGEKGPDC (C-iRGE) with no $\alpha\beta3/\alpha\beta5$ binding capacity, which has been similarly attached on T cell surface. Besides, a function-blocking anti-NRP1 antibody and control anti-IgG antibody have also been used to inhibit the tumor penetration of T-iRGD. Tumor specific penetration capacity of additional three groups have been tested in 3D tumor spheroids as well as mouse peritoneal metastatic gastric tumor model. The results are shown in **Figure 2a** and **Extended Data Figure 5**. The legend for **Figure 2a** and **Extended Data Figure 5** has also been revised.

Figure 2a. Confocal microscopy images and surface plot images of HGC27 spheroids treated with CFSE labeled T, T+iRGD, T-iRGD or control peptide iRGE modified T cells (T-iRGE) for 20 hours. 15ug/ml anti-NRP1 antibody and control sheep anti-IgG were added 20min before T cells incubation. Magnification is 200x; Scale bars, 100 μ m.

Extended Data Figure 5 iRGD-mediated in vivo tumor penetration of T cells in intraperitoneal tumor model. Mice bearing intraperitoneal tumor were treated with 50ug anti-NRP1 antibody or control sheep anti-IgG 15 min prior to T cells injection, tumor nodules were analyzed 3h post T cells injection.

We have revised the information of **result** as follows:

Strikingly, iRGD modification resulted in lymphocytes being able to infiltrate deep into the MCS core in 20 hours, nevertheless preinjection of an antibody to NRP1 (anti-NRP1) inhibited the iRGD-induced T cells penetration. The control peptide iRGE modified T cells also showed no tumor penetrating capacity (Fig. 2a). These results indicated that iRGD enhanced tumor specific T cells accumulation is both RGD and CendR motif dependent.

Near-infrared imaging demonstrated that only the T-iRGD group exhibited the early accumulation of T cells (3 hours post-injection) in small tumor nodules (Fig. 2d-f and Extended Data Fig. 5).

We have added the information of **method** as follows:

To study the MCS penetration of T lymphocytes, T cells were stained with 4 μ M CFSE (Carboxyfluorescein succinimidyl ester) (Abcam, UK) for 10 minutes at 37 $^{\circ}$ C in PBS before iRGD modification. Labeling was stopped by 5-fold volume of cold complete medium (10% FBS in AIM-V) and extensively washing for three times. Established spheroids were firstly incubated with 15ug/ml function-blocking anti-NRP1 antibody (R&D Systems) or control sheep anti-IgG (R&D Systems) for 20min at 4 $^{\circ}$ C. CFSE labeled T cells, T cells combined with iRGD, T cells modified with DSPE-PEG-iRGD or T cells modified with DSPE-PEG-iRGE were then added to the solution and incubated with the spheroid for 20 hours.

To investigate the tumor targeting efficiency of T-iRGD in tumor-bearing mice, 10^7 T cells stained with near-infrared fluorescent probe DiR (Bridgen, China) were injected intraperitoneally (MKN45 peritoneal metastasis mouse model) or intravenously (HGC27 and SNU719 subcutaneous mouse model). At different time intervals, the mice were anesthetized and scanned using a CRi MaestroTM Automated In Vivo Imaging System (C.R. INTERNATIONAL INC, USA). In some experiments, 50 ug of function-blocking anti-NRP1 antibody or sheep IgG was intravenously injected into the tumor mice 15 min prior to the T cells injections.

Comment 2:

The last section on the mechanism of iRGD-induced T cell penetration is woefully incomplete and fails to adequately support the conclusion presented in the title of the section. Minimally, experiments such as inhibiting the phosphorylation with anti-NRP1 and using endothelial cells with a mutant V-cadherin that cannot be phosphorylated would be needed. Direct visualization of lymphocyte penetration through tumor endothelium by EM, as was done in reference 23 with nanoparticles, would be another way of demonstrating the authors claim.

Answer:

Thank you very much for your comments. To make our experiments more rigorous, we have added three groups: inhibiting the binding of iRGD to HUVECs with anti-NRP1 or control anti-IgG antibody and inhibiting the phosphorylation of VE-cadherin with a PI3K γ/δ inhibitor TG100-115. And we have added the result in **Figure 4c**:

Figure 4c. Anti-NRP1 antibody and PI3K γ/δ inhibitor TG100-115 blocked iRGD induced tyrosine phosphorylation of VE-cadherin. Endothelial cells were incubated with 15ug/ml anti-NRP1 antibody, control sheep anti-IgG or 3.5ug/ml TG100-115 prior to iRGD incubation, tyrosine phosphorylation of VE-cadherin was analyzed by immunoprecipitation.

We have improved the last section "**The enhanced tumor-specific infiltration of iRGD-modified T cells depends on the tyrosine phosphorylation of VE-cadherin**" as follows:

Numerous studies have demonstrated that the vascular endothelial adhesion molecule, VE-cadherin, regulates the stability of endothelial junctions and participates in the process of lymphocyte transendothelial migration[2, 3]. Tyrosine phosphorylation of

VE-cadherin has been demonstrated to induce the disengagement of endothelial junctions and the formation of intercellular gaps, and thus, transendothelial lymphocyte migration[3]. For NRP-1 activation has been reported to induce tyrosine phosphorylation of VE-cadherin in a PI3K γ / δ dependent manner[4], we aimed to understand whether the mechanism also plays a role in iRGD induced lymphocyte infiltration. In endothelial cells, iRGD treatment led to a rapid and transient increase in VE-cadherin tyrosine phosphorylation (Fig. 4a-b). To extend these studies, endothelium cells were pretreated with a function-blocking NRP-1 antibody or the PI3K γ / δ inhibitor TG100-115 before iRGD incubation. As expected, both NRP-1 antibody and PI3K γ / δ inhibitor potently inhibited iRGD-induced tyrosine phosphorylation of VE-cadherin. This research verified the underlying mechanism of iRGD in promoting lymphocyte infiltration that the binding of iRGD to NRP-1 triggered the formation of intercellular gaps through the tyrosine phosphorylation of VE-cadherin (Fig. 4c).

We have added the method in **Immunoprecipitation of VE-cadherin.**

Confluent HUVECs (p 4-6) were starved in serum-free EBM-2 media (Lonza, USA) for 16 hours prior to 20 minutes incubation with 15ug/ml anti-NRP1 antibody, control sheep anti-IgG or 3.5ug/ml TG100-115. Then, cells were stimulated with iRGD (500 ng/mL) for 15-minute or 30-minute. Immunoprecipitation of VE-cadherin was performed as described previously[4].

Comment 3:

There are some writing-related issues that could cause confusion:

‘Aggregation’ is used in the meaning of ‘accumulation’ in describing accumulation of T cells in tumors. As aggregation generally means clustering in the context of cells, this usage is confusing.

“Immunohistochemical analysis confirmed the intensive tumor-specific infiltration of iRGD-modified lymphocytes: CD3+ T cells were found deep inside large peritoneal tumor nodules ...” A reader wonders why T cells now became CD3+ T cells. The figure legend explains that the antibody used to detect T cells was anti-CD3. “T cells, as detected by CD3 antibodies” would take care of it. ‘Systematically’ should be ‘systemically’, ‘merely’ ‘only’, and ‘cycled’ ‘cyclic.

Answer:

Thank you very much for your corrections. We have replaced “aggregation” with “accumulation” in describing accumulation of T cells in tumors.

We have rephrased the sentence as follows:

Immunohistochemical analysis confirmed the intensive tumor-specific infiltration of iRGD-modified lymphocytes: T cells, as detected by CD3 antibodies were found deep

inside large peritoneal tumor nodules while the other two groups just showed T cells in the tumor periphery (Extended Data Fig. 6c).

We have corrected all of the mentioned spelling mistakes.

Reviewer #2:

Comment 1:

1. The T-cell accumulation data shown in Fig 3a-c is intriguing. However, analysis is stopped after 6 days. Please provide longer follow up.

Answer:

Thanks for your comments. We applied different scale for each time point, which makes our results confusing. To show our results in a clearer manner, we have unified the scale in **Figure 3a**, which is also suggested by reviewer 3. In this case, T cell signal is attenuated with time and the overall signal is relatively low on day 6 in all groups, so we didn't extend the observation after 144 hours.

Figure 3a In vivo imaging of SNU719 tumor-bearing mice at 6, 24, 72 and 144 h after intravenous injection of DiR labeled T cells. White dashed lines, subcutaneous tumors.

We designed the experiment mainly to test the tumor-target infiltration capacity of iRGD modified T cell in an intuitive way. Pittet et al. monitored T cells distribution for 120 hours and showed that as early as 2 h after transfer, the adoptively transferred T cells begin to accumulated in the tumors [5]. Santos et al showed that at day 2 and day 3, T cell signal is apparent in tumor area[6]. Youniss et al. also showed that T cell signal in tumor area peaked on day 3 after adoptive transferring [7]. So 48 hour to 72 hour may be the best time to monitor the tumor target infiltration of lymphocytes [5, 7]. In our experiment, we could see that on day 1 and day 3, the iRGD modification group showed a distinct T cell signal in tumor area whereas in T cell group, no T cell

signal was shown. From this result, we could draw the conclusion "**iRGD modification enhances the infiltration of T cells into subcutaneous tumors**".

Except for tumor specific infiltration, T cell biodistribution in normal organs is another important parameter. Studies showed that vast majority of transferred T cells accumulate in the lungs 2 h post injection but rapidly redistribute to the liver and spleen within 24 h and the signal attenuate with time[8]. In three groups we have tested, no T cell signal was seen in normal organs except for liver and spleen. Besides, the T cells signal in normal organs is attenuated with time. On day 6, the T cell signal in normal organs is already low so we didn't extend our observation after day 6. These results could rule out concerns on non-specific infiltration of normal tissues caused by iRGD modification.

Comment 2:

The animal experiments testing the antitumor activity of iRGD-loaded T cells are confounded by the administration of multiple doses of very high doses of IL-2. In this regard, it is critical to have a group of mice as controls that only receive IL-2.

Answer

Thanks for your comments. Adoptively transferred T cells do not survive in vitro or in vivo for long in the absence of IL-2, so the administration of high doses of IL-2 is accompanied by T cells transfer to prolong the survival of activated T cells in vivo[9-12]. This therapeutic regimen could also be seen in clinical trials of adoptive T cell immunotherapy[13, 14]. Besides, we applied human IL-2, which may have no impact on mouse models with no humanized IL-2 receptor expression, such as the immunodeficient mouse model we adopted in present study.

The grouping method and the dose of administered IL-2 are in reference to several researches with some modifications[9, 11, 12]. For example, when test the effect of local irradiation on adoptive T cell immunotherapy, Wei et al.[12] designed experiment as follows: "Mice were treated either with daily local irradiation at 8.5 Gy between day 7 and 11 (RT); or adoptively transferred 30×10^6 ex-vivo activated TDLN cells on day 11 (AT); or combined local radiation and T cell therapy (AT+RT). Following T cell transfer, mice received 40,000 IU IL-2 by i.p. bid, for a total of 8 doses."

Comment 3:

It is unclear why the investigators use unspecific T cells for the MKN45 experiments. In this regard please provide data that these allogeneic T cells recognize and kill MKN45 cells (right now only growth inhibition of spheroids is shown in Fig 2c). Also, for the 2nd model, please provide data that the injected LMP2-specific T cells recognize and kill SUN719 cells ex vivo; while the authors have published this before, generating these cells is in part donor dependent. Lastly, I could not find the HLA type of SUN719 cells in the literature-please provide.

Answer

Thank you very much for your comments. MKN45 is one of the commonly used cell lines for the construction of peritoneal tumor model of gastric cancer[15, 16]. We choose unspecific T cells for the MKN45 experiments for two reasons: Firstly, the infiltration of antigen specific T cells to tumor site requires the expression of tumor-specific cognate antigen[5, 6, 17]. We aimed to rule out the interference from antigen recognition when evaluating the function of iRGD in tumor target T cells infiltration.

Secondly, we adopted antigen specific T cells in the second model and we aimed to testify the function of iRGD in different kind of T cells. Besides, the iRGD modification strategy is universal, which means all kinds of T cells can be targeted into solid tumor tissues. Needless to say, further studies to investigate the usage of iRGD in CAR-T and TCR-T adoptively transfer are better ways to validate the potential role of iRGD.

We have tested unspecific T cells killing using CFSE/PI labeling cytotoxicity assay and showed that allogeneic T cells could recognize and kill MKN45 cells in vitro. Besides, no significant difference was seen in T cell group, T+iRGD group or T-iRGD group.

We have added the result in **Extended Date Figure 7**.

Extended Data Figure 7 Cytotoxicity of iRGD modified T cell on MKN45 in vitro. Activated T cells of different format were incubated with CFSE labeled MKN45 cells at effector-to-target ratio (E:T) of 5:1,10:1,20:1 and 40:1 respectively, PI was added 8

hours after incubation and the percentage of dead cells was analyzed by flow cytometry.

The ex vivo killing assay of LMP2-specific T cells also showed that LMP2-specific T cells could recognize and kill SNU719 in vitro and iRGD modification didn't compromise T cell killing capacity. This result was shown in **Extended Data Figure 2c**.

The HLA type of SNU719 cells is HLA-A*24:02, which has also been reported in website <http://cellines.tron-mainz.de/>.

Type	Allele 1	Allele 2	RPKM	Information	Source
A	24:02	24:02	178.47		seq2HLA
B	07:02	52:01	35.82		seq2HLA
C	12:02	07:02	36.72		seq2HLA
DQA1	05:02*	05:02	0.08	Ambiguity on 4 digit level. Chosen allele had the highest number of reads	seq2HLA
DQB1	05:01	05:01	5.03		seq2HLA
DRB1	15:02*	01:01	4.71	Ambiguity on 4 digit level. Chosen allele had the highest number of reads	seq2HLA

We have added this information in **Methods** as follows:

For subcutaneous tumor model, 6-8-week-old male BALB/c nude mice were injected subcutaneously with HLA-A24 positive SNU719 cells (1×10^7 suspended in 100 ul PBS) or HGC27 cells (5×10^6 suspended in 100 ul PBS).

The information of HLA type of SNU719 and corresponding HLA-A24-restricted LMP2A peptide has also been added in the figure legend of **Extended Data Figure 2: c-e**. *Cytotoxicity, cellular response and phenotype of EBV-LMP2A-specific CTLs of different format. LMP2A-specific CTLs were induced by co-culturing of HLA-A24 positive human PBMC with HLA-A24-restricted LMP2A peptide (TYGPVFMCL) loaded DCs in the presence of IL-2, IL-7 and IL-15 for 14 days. And then, EBV-LMP2A-CTLs were incubated with iRGD or DSPE-PEG-iRGD solutions at the concentration of 5 μ g/ml for 30 minutes and analyzed. LMP2A-specific CTLs of different format were incubated with CFSE labeled HLA-A24 positive SNU-719 cells at effector-to-target ratio (E:T) of 5:1, 10:1, 20:1 and 40:1 respectively, PI was added 6 hours after incubation and the percentage of dead cells was analyzed by flow cytometry.*

Comment 4

No complete tumor regressions were observed in the animal models. In addition, the shown survival benefit in the 2nd animal model seems marginal (no stat analysis was provided). Thus, believe that it is critical that the authors provide robust mechanistic studies. The studies shown in Fig 4 have to be considered preliminary.

Answer

Thanks for very much for your remark.

The *in vivo* antitumor process of adoptively transferred T cells is complicated and determined by several factors. Sufficient T cells infiltration is one of the prerequisites

for effective anti-tumor effect. Factors such as the inhibitive microenvironment, T cell type, T cell vitality and life span are all essential to tumor control. As we have answered in question 3, the T cells we adopted are comparatively weak in cell killing capacity. Even in this case, though, the inhibitory rate of T-iRGD is superior in both peritoneal tumor model and subcutaneous tumor model. In the 2nd animal model, the inhibitory rate of T-iRGD in combination with PD-1 disruption reach to 74.7%, compared to 27.3% in PD-1 disruption group. Besides, iRGD modification also increased the survival rate (shown in survival curve, PD1-KO-CTL v.s iRGD-PD1-KO-CTL, $p=0.01$). We believe that iRGD modification group may have better performance when using T cells with superior killing capacity.

For the mechanistic studies, we have refined our research work as proposed by reviewer 1.

Comment 5

Please provide stat analysis for the survival curve shown in Fig 3f.

Answer

Thank you very much for your comments. We have analyzed the survival curve with log-rank test and marked p value in **Fig. 3f**

Reviewer #3

Comment 1

The whole-body scan images do not contain appropriate scale, which needs to be fixed by the authors. The authors have presented just a color ribbon, with no numerical reference to it.

Answer

Thanks for your suggestion. We applied different scale for each time point, which makes our results confusing. To display the result more clearly, we modified the presentation form of our results and put different groups in the same overall scale and labeled the scale number in **Figure 2e, Figure 3a, Extended Data Figure 5, Extended Data Figure 6a and Extended Data Figure 9a.**

Comment 2

In live animal imaging studies (Fig 3a and ExtendedData7a) one can see that as time passes (from 6h -> 144h), the overall signal in the whole animal scan is increasing with time, while one would assume that the signal decreases in time. Are the different time points (6h vs 24h vs 72h vs 144h) in the same overall scale? Or does a different scale apply for each time point? How do the authors explain that as time passes, the amount of their injected material (T-cells injected at time 0h) is increasing? In the ExtendedData7a "T-iRGD" group it even seems that the T-cells are labelling up the whole body (including the head region, which the authors do not comment).

Answer

Thanks for your comments. As we have answered in comment 1, we have unified the scale of different groups in **Figure 3a and Extended Data Figure 9a**. In this case, the overall signal in the whole animal scan is decreasing with time.

To avoid signal interference from liver and spleen, we injected tumor cell suspensions on the mice neck subcutaneous part. In this case, the signal in tumors looks like in head region. To make a clearer exhibition, we circled tumor area using white dashed lines as follows:

Figure 3a In vivo imaging of SNU719 tumor-bearing mice at 6, 24, 72 and 144 h after intravenous injection of DiR labeled T cells. White dashed lines, subcutaneous tumors.

The overall signal intensity in our experiment is relatively low so we could see non-specific background fluorescent signal on skin surface, which doesn't represent the T cell distribution. This phenomenon could also be seen in some other publications[8, 18, 19].

Comment 3

Fig 3a live animal scans in T-iRGD group contain a strong signal origin that the authors do not comment on. It is impossible for the reader to interpret the signal (the images are small and of low quality), but it seems that one side/part of the lungs gets infiltrated with the lymphocytes? Similarly, as mentioned before, in ExtendedData7a at 144h one can see very strong signal in the head region.

Answer

Thanks for your comments. As we have answered in comment 1 and comment 2, when applied the same scale for different time point, no signal could be seen in normal organs except for liver, spleen, tumor and some skin background fluorescence.

Comment 4

Minor and technical remarks

Language

- L99 one instance of "lymphocytes" should be removed.
- In the whole manuscript, there are several instances where "systematic injection" should be replaced with "systemic injection".

Illegible Figures

- There are illegible (too small) elements in several figures: for example, Fig 1b, Fig 2h.
- Extended Fig 1a – the top is missing

Answer

Thank you very much for the remind. We have corrected the mentioned spelling mistakes and articulated our figures as shown in **Figure 1b, Figure 2h and Extended Figure 1a**.

Comment 5

Replicability: the authors should make sure their study can be replicated by the other scientists:

- The monitoring and measurements of MCS (spheres). L406 "MCS ... were used for functional assays upon reaching a solid state approximately 72 hours" – the size of the spheroids should be indicated.
- When presenting the results of the infiltration of labelled T-cells into the MCS, L414 "...T cells in MCS were monitored using confocal microscope", but the authors do not mention or describe how they ensured the comparability between the different spheroids. Usually the imaging and quantitation is performed at the mid-height of the equal-sized spheroids.

Answer

Thanks for your remind and we have revised the **Materials and Methods** part as follows:

The HGC27 cells (500 in 150 ul of complete media) were added to 96 Well Clear Round Bottom Ultra Low Attachment Microplate (Corning, USA) and allowed to grow up at 37 °C to attain the diameter about 200µm for 72 hours.

MCS were monitored with a microscope and the uniform and compact tumour spheroids were selected for the subsequent studies. To study the MCS penetration of T lymphocytes, T cells were stained with 4 µM CFSE (Carboxyfluorescein succinimidyl ester) (Abcam, UK) for 10 minutes at 37 °C in PBS before iRGD modification. Labeling was stopped by 5-fold volume of cold complete medium (10% FBS in AIM-V) and extensively washing for three times. Established spheroids were first incubated with 15µg/ml function-blocking anti-NRP1 antibody (R&D systems, AF3870) or control sheep anti-IgG (R&D systems, 5-001-A) for 20min at 4 °C. CFSE labeled T

cells, T cells combined with iRGD, T cells modified with DSPE-PEG-iRGD or T cells modified with DSPE-PEG-iRGE were then added to the solution and incubated with the spheroid for 20 hours. After washing and fixing in 4% paraformaldehyde, tumour spheroids were scanned from the top to the middle with 10 µm intervals using confocal microscope (ZEISS, Germany) and the images were acquired at the mid-height of the spheroid.

We are submitting a clean copy of the manuscript and a copy with the edits tracked. We will be happy to be of assistance during the review process.

Yours sincerely,

Baorui Liu, MD, PhD
Jia Wei, MD, PhD
The Comprehensive Cancer Centre of Drum Tower Hospital
Medical School of Nanjing University
Clinical Cancer Institute of Nanjing University
Nanjing, China

References

1. Sugahara, K.N., et al., *Tissue-penetrating delivery of compounds and nanoparticles into tumors*. *Cancer Cell*, 2009. **16**(6): p. 510-20.
2. Nourshargh, S. and R. Alon, *Leukocyte migration into inflamed tissues*. *Immunity*, 2014. **41**(5): p. 694-707.
3. Turowski, P., et al., *Phosphorylation of vascular endothelial cadherin controls lymphocyte emigration*. *J Cell Sci*, 2008. **121**(Pt 1): p. 29-37.
4. Acevedo, L.M., et al., *Semaphorin 3A suppresses VEGF-mediated angiogenesis yet acts as a vascular permeability factor*. *Blood*, 2008. **111**(5): p. 2674-80.
5. Mikael J. Pittet*†‡, J.G., Cedric R. Berger†, Takahiko Tamura†, Gregory Wojtkiewicz†, Matthias Nahrendorf†, Pedro Romero¶, Filip K. Swirski†, and Ralph Weissleder*†, *In vivo imaging of T cell delivery to tumors after adoptive transfer therapy*. *PNAS*, 2007. **vol. 104**
6. Santos, E.B., et al., *Sensitive in vivo imaging of T cells using a membrane-bound Gaussia princeps luciferase*. *Nat Med*, 2009. **15**(3): p. 338-44.
7. Bachmann, M.P., et al., *Near-Infrared Imaging of Adoptive Immune Cell Therapy in Breast Cancer Model Using Cell Membrane Labeling*. *PLoS ONE*, 2014. **9**(10): p. e109162.
8. Caruana, I., et al., *Heparanase promotes tumor infiltration and antitumor activity of CAR-redirection T lymphocytes*. *Nat Med*, 2015. **21**(5): p. 524-9.
9. Gordon-Alonso, M., et al., *Galectin-3 captures interferon-gamma in the tumor matrix reducing chemokine gradient production and T-cell tumor infiltration*. *Nat Commun*, 2017. **8**(1): p. 793.

10. Shrikant, P. and M.F. Mescher, *Opposing effects of IL-2 in tumor immunotherapy: promoting CD8 T cell growth and inducing apoptosis*. J Immunol, 2002. **169**(4): p. 1753-9.
11. Teitz-Tennenbaum, S., et al., *Radiotherapy combined with intratumoral dendritic cell vaccination enhances the therapeutic efficacy of adoptive T-cell transfer*. J Immunother, 2009. **32**(6): p. 602-12.
12. Wei, S., et al., *Effects of tumor irradiation on host T-regulatory cells and systemic immunity in the context of adoptive T-cell therapy in mice*. J Immunother, 2013. **36**(2): p. 124-32.
13. Chandran, S.S., et al., *Treatment of metastatic uveal melanoma with adoptive transfer of tumour-infiltrating lymphocytes: a single-centre, two-stage, single-arm, phase 2 study*. Lancet Oncol, 2017. **18**(6): p. 792-802.
14. Lee, J.H., et al., *Adjuvant immunotherapy with autologous cytokine-induced killer cells for hepatocellular carcinoma*. Gastroenterology, 2015. **148**(7): p. 1383-91 e6.
15. Simon-Gracia, L., et al., *iRGD peptide conjugation potentiates intraperitoneal tumor delivery of paclitaxel with polymersomes*. Biomaterials, 2016. **104**: p. 247-257.
16. Sugahara, K.N., et al., *A tumor-penetrating peptide enhances circulation-independent targeting of peritoneal carcinomatosis*. J Control Release, 2015. **212**: p. 59-69.
17. Boissonnas, A., et al., *In vivo imaging of cytotoxic T cell infiltration and elimination of a solid tumor*. J Exp Med, 2007. **204**(2): p. 345-56.
18. Xue, J., et al., *Neutrophil-mediated anticancer drug delivery for suppression of postoperative malignant glioma recurrence*. Nat Nanotechnol, 2017. **12**(7): p. 692-700.
19. Song, X., et al., *Development of a multi-target peptide for potentiating chemotherapy by modulating tumor microenvironment*. Biomaterials, 2016. **108**: p. 44-56.

Reviewers' Comments:

Reviewer #2:

Remarks to the Author:

In the submitted manuscript, the authors demonstrate that loading the tumor penetrating cyclic peptide iRGD on the cell surface of T cells enhances their ability to penetrate tumors in vitro and in vivo. The revised manuscript is overall improved, but I continue to have several concerns.

Major Concerns:

1) I am still concerned that all in vivo experiments were conducted with high dose IL2; the authors state in their rebuttal comment to reviewer 2, comment 2, that human T cells do not survive in immunodeficient mice without IL2. This is not correct; numerous adoptive T-cell transfer studies have been conducted in immunodeficient mice without the addition of IL2 (for example, please review CAR T-cell literature). While some investigators have used IL2 (for example for studying EBV-specific CTLs in xenograft models), the IL2 concentration was one log lower (~3,000 units per dose ip). Lastly, the authors state that human IL2 does not activate murine IL2 receptors. This is not correct, human IL2 is crossreactive with murine IL2.

2) I am continue to be puzzled by the fact that the authors decided to use a tumor model with T cells that 'are comparatively weak in cell killing capacity (quote from the authors' rebuttal letter)'. For example, if T cells with better cell killing capacity could overcome initial fewer T-cell numbers at tumor sites by proliferation, then the developed approach by the authors would have a limited impact.

Minor concerns

1) While the authors now provide evidence of direct tumor cell killing by T cells (Figure 7), the experiments lacks controls (e.g. incubation of tumor cells without T cells). Re-review of the extended data figure 2, revealed also lack of controls for panels 2c (unspecific T cells as effectors), and 2d (unspecific T cells as effectors and no target cells for all conditions).

Reviewer #3:

Remarks to the Author:

the manuscript is ready for publication now

RE: NCOMMS-18-01750A "iRGD synergizes with PD-1 knockout immunotherapy by enhancing lymphocyte infiltration in gastric cancer"

Dear Professor Baratta :

Thank you for your email on 24 October, with the reviewers' additional comments on our referenced manuscript. We have revised the manuscript again in accordance with their comments, as follows:

Reviewer #2:

Comment 1:

I am still concerned that all in vivo experiments were conducted with high dose IL2; the authors state in their rebuttal comment to reviewer 2, comment 2, that human T cells do not survive in immunodeficient mice without IL2. This is not correct; numerous adoptive T-cell transfer studies have been conducted in immunodeficient mice without the addition of IL2 (for example, please review CAR T-cell literature). While some investigators have used IL2 (for example for studying EBV-specific CTLs in xenograft models), the IL2 concentration was one log lower (~3,000 units per dose ip). Lastly, the authors state that human IL2 does not activate murine IL2 receptors. This is not correct, human IL2 is crossreactive with murine IL2.

Answer

Thank you very much for your comments. The dose of IL-2 administration is in reference to several researches¹⁻⁴. For example, Su et al.⁴ gave 40,000U IL-2 by i.p. once a day for three consecutive days after T cell transfer. Wei et al.² gave 40,000 IU IL-2 by i.p. bid, for a total of 8 doses and Teitz-Tennenbaum¹ gave 42,000 IU IL-2 bid, for a total of 8 doses.

To make our experiments more rigorous, we have added a group that only receive IL-2 as controls and the result was shown in **Figure 2i, 2j, Figure 3e and Figure 3g**. According to these figures, there is no anti-tumor effect in IL-2 group.

Figure 2 T-iRGD showed improved tumor infiltration capacity and anti-tumor efficiency in 3D tumor spheroids and peritoneal metastasis tumor model.

i-j. Mice bearing disseminated MKN45 peritoneal tumors implanted 1 week earlier received intraperitoneal injection of PBS, IL-2, T, T+iRGD or T-iRGD every 4 days for 3 times. Tumors are harvested after 2 weeks of treatment. Tumor nodules larger than 3 mm in diameter were weighed (i), and the numbers of remaining small tumor nodules (1-3 mm in general) were counted (j).

Figure 3 iRGD modification enhanced lymphocytes infiltration into tumor parenchyma in a systematic administration route and overcame resistance to PD-1 disruption immunotherapy.

*Enhanced antitumor effect of iRGD modified PD-1-disrupted CTLs in a xenograft SNU719 mouse gastric tumor model. Tumor-bearing mice received different forms of treatment every 4 days for 3 times. Tumor growth profiles (e) of mice treated with PBS, IL-2, CTL, CTL+iRGD, CTL-iRGD, PD-KO-CTL, PD1-KO-CTL+iRGD and PD1-KO-CTL-iRGD. Weight of tumors collected 2 weeks post treatment (g). Survival curves were analyzed with log-rank test. Tumor volume and tumor weight were analyzed with Student's t test. Data are represented as mean \pm s.e.m., n = 7; n.s, not significant; *p < 0.05; **p < 0.01; ***p < 0.001.*

Comment 2:

I am continuing to be puzzled by the fact that the authors decided to use a tumor model with T cells that 'are comparatively weak in cell killing capacity (quote from the authors' rebuttal letter)'. For example, if T cells with better cell killing capacity could overcome initial fewer T-cell numbers at tumor sites by proliferation, then the developed approach by the authors would have a limited impact.

Answer

Thank you very much for your comments. Although better cell killing capacity is fundamental for the success of immunotherapy, T cells with better cell killing capacity such as CAR-T are still hampered by the tumor infiltration deficiency, especially in the case of solid tumors⁵⁻⁷. Ignazio Caruana found that *in vitro*-cultured T lymphocytes are defective in their capacity to degrade the ECM, which result in the reduced capacity of cultured CAR-T cells to penetrate stroma-rich solid tumors. To solve this problem, they engineered CAR-T cells to express HPSE to degrade the ECM and mice infused with high HPSE expression CAR-T had significantly improved survival compared to mice treated with control⁸. Di Stasi, A. *et al.* also demonstrated that active T cells express low levels of a chemokine receptor CCR4. Genetic modification of CAR-T with CCR4 can substantially improve their homing and antitumor activity *in vivo*⁹. So, increasing efforts has been made to improve T cells infiltration into the tumor parenchyma. In this case, our research is of great value, because of the simple process, low cost, less time consuming and high universality of T cells modification procedures. We provide a universal modification strategy not only for the T cells used in present study, but also for CAR-T and TCR-T therapy.

Comment 3:

While the authors now provide evidence of direct tumor cell killing by T cells (Figure 7), the experiments lacks controls (e.g. incubation of tumor cells without T cells). Re-review of the extended data figure 2, revealed also lack of controls for panels 2c and 2d (unspecific T cells as effectors and no target cells for all conditions).

Answer

Thanks for your comments. We have supplemented the result of controls in **Supplementary Figure 7, Supplementary Figure 2c and 2d**. Supplementary Figure 7 and Supplementary Figure 2c showed the ratio of dead tumor cells without T cells incubation. Supplementary Figure 2d showed the IFN- γ level in culture supernatant without target cells.

Supplementary Figure 7 Cytotoxicity of iRGD modified T cell on MKN45 *in vitro*. Activated T cells of different format were incubated with CFSE labeled MKN45 cells at effector-to-target ratio (E:T) of 0:1, 5:1, 10:1, 20:1 and 40:1 respectively, PI was added 6 hours after incubation and the percentage of dead cells was analyzed by flow cytometry.

Supplementary Figure 2 Influence of DSPE-PEG-based modification on T cells *in vitro*. **c-d** Cytotoxicity, cellular response and phenotype of EBV-LMP2A-specific CTLs of different format. LMP2A-specific CTLs were induced by co-culturing of HLA-A24 positive human PBMC with HLA-A24-restricted LMP2A peptide (TYGPVFMCL) loaded DCs in the presence of IL-2, IL-7 and IL-15 for 14 days. And then, EBV-LMP2A-CTLs were incubated with iRGD or DSPE-PEG-iRGD solutions at the concentration of 5 $\mu\text{g ml}^{-1}$ for 30 minutes and analyzed. LMP2A-specific CTLs of different format were incubated with CFSE labeled HLA-A24 positive SNU719 cells at effector-to-target ratio (E:T) of 0:1, 5:1,10:1,20:1 and 40:1 respectively, PI was added 6 hours after incubation and the percentage of dead cells was analyzed by flow cytometry (c). CTLs were stimulated with LAM2A loaded autologous DCs for 20 hours, culture supernatant was collected and the cytokine level was analyzed by CBA Human IFN- γ kit (d).

参考文献

- 1 Teitz-Tennenbaum, S. *et al.* Radiotherapy combined with intratumoral dendritic cell vaccination enhances the therapeutic efficacy of adoptive T-cell transfer. *J Immunother* **32**, 602-612,(2009).
- 2 Wei, S., Egenti, M. U., Teitz-Tennenbaum, S., Zou, W. & Chang, A. E. Effects of tumor irradiation on host T-regulatory cells and systemic immunity in the context of adoptive T-cell therapy in mice. *J Immunother* **36**, 124-132,(2013).
- 3 Gordon-Alonso, M., Hirsch, T., Wildmann, C. & van der Bruggen, P. Galectin-3 captures interferon-gamma in the tumor matrix reducing chemokine gradient production and T-cell tumor infiltration. *Nat Commun* **8**, 793,(2017).
- 4 Su, S. *et al.* CRISPR-Cas9-mediated disruption of PD-1 on human T cells for adoptive cellular therapies of EBV positive gastric cancer. *Oncoimmunology* **6**, e1249558,(2017).
- 5 Zheng, H. *et al.* HDAC Inhibitors Enhance T-Cell Chemokine Expression and Augment Response to PD-1 Immunotherapy in Lung Adenocarcinoma. *Clin Cancer Res* **22**, 4119-4132,(2016).
- 6 Sackstein, R., Schatton, T. & Barthel, S. R. T-lymphocyte homing: an underappreciated yet critical hurdle for successful cancer immunotherapy. *Lab Invest* **97**, 669-697,(2017).
- 7 Tang, H. *et al.* Facilitating T Cell Infiltration in Tumor Microenvironment Overcomes Resistance to PD-L1 Blockade. *Cancer Cell* **29**, 285-296,(2016).
- 8 Caruana, I. *et al.* Heparanase promotes tumor infiltration and antitumor activity of CAR-redirected T lymphocytes. *Nat Med* **21**, 524-529,(2015).
- 9 Di Stasi, A. *et al.* T lymphocytes coexpressing CCR4 and a chimeric antigen receptor targeting CD30 have improved homing and antitumor activity in a Hodgkin tumor model. *Blood* **113**, 6392-6402,(2009).

Reviewers' Comments:

Reviewer #2:

Remarks to the Author:

The authors have addressed all my concerns with the revised version of the manuscript.

Minor comment:

Please provide a figure legend for Supplementary Figure 3; right now only a title is provided; also the same FITC isotype control is used in the middle and right panel (that is of course OK; recommend that this is stated in the legend to prevent future queries).

RE: NCOMMS-18-01750C "iRGD synergizes with PD-1 knockout immunotherapy by enhancing lymphocyte infiltration in gastric cancer"

Dear Professor Baratta :

Thank you for your email on 13 February, with the reviewers' additional comments on our referenced manuscript. We have revised the manuscript again in accordance with their comments, as follows:

Reviewer #2:

Comment 1:

Please provide a figure legend for Supplementary Figure 3.

Answer

Thank you very much for your comments. We have added a figure legend in **Supplementary Figure 3.**

Supplementary Figure 3 Cell expression of $\alpha\beta 3$, $\alpha\beta 5$ and NRP-1.

$\alpha\beta 3$, anti- $\alpha\beta 5$ and NRP-1 expression in tumor cells was determined by flow cytometry. Integrin $\alpha\beta 3$ was detected using FITC-conjugated mouse anti-human $\alpha\beta 3$ monoclonal antibody, and integrin $\alpha\beta 5$ was detected using FITC-conjugated mouse anti-human $\alpha\beta 5$ monoclonal antibody. The matched isotype control was FITC-conjugated mouse IgG1 κ . NRP-1 was detected using PE-conjugated mouse anti-human NRP-1 monoclonal antibody and an isotype control.

We will be happy to be of assistance during the review process.